# A Lunar-Orbiting Satellite Constellation for Wireless Energy Supply

Francesco Lopez [1], Anna Mauro [2], Stefano Mauro [1,*], Giuseppe Monteleone [2], Domenico Edoardo Sfasciamuro [1] and Andrea Villa [2]

---

[1] Department of Mechanical and Aerospace Engineering, Politecnico di Torino, 10129 Torino, Italy; francesco.lopez@polito.it (F.L.); domenico.sfasciamuro@polito.it (D.E.S.)

[2] Department of Electronics and Telecommunications, Politecnico di Torino, 10129 Torino, Italy; anna.mauro@polito.it (A.M.); giuseppe.monteleone@polito.it (G.M.); andrea.villa@polito.it (A.V.)

\* Correspondence: stefano.mauro@polito.it

**Abstract:** The goal of this research is to define a lunar-orbiting system that provides power to the lunar surface through wireless power transmission. To meet the power demand of a lunar base, a constellation of satellites placed in stable orbits is used. Each satellite of this constellation consists of solar arrays and batteries that supply a power transmission system. This system is composed of a laser that transmits power to receivers on the lunar surface. The receivers are photonic power converters, photovoltaic cells optimized for the laser's monochromatic light. The outputs of this work will cover the architecture of the system by studying different orbits, specifically analyzing some subsystems such as the laser, the battery pack and the receiver placed on the lunar ground. The study is conducted considering two different energy demands and thus two different receivers location: first, at the strategic location of the Artemis missions' landing site, the Shackleton Crater near the lunar south pole; second, on the lunar equator, in anticipation of future and new explorations. The goal is to evaluate the possible configurations to satisfy the power required for a lunar base, estimated at approximately 100 kW. To do this, several cases were analyzed: three different orbits, one polar, one frozen and one equatorial (Earth–Moon distant retrograde orbit) with different numbers of satellites and different angles of the receiver's cone of transmission. The main objective of this paper is to perform a comprehensive feasibility study of the aforementioned system, with specific emphasis placed on selected subsystems. While thermal control, laser targeting, and attitude control subsystems are briefly introduced and discussed, further investigation is required to delve deeper into these areas and gain a more comprehensive understanding of their implementation and performance within the system.

**Keywords:** wireless power transmission; Moon energy supply; laser; frozen orbit; Earth–Moon distant retrograde orbit

## 1. Introduction

The primary focus of this research is to tackle the issue of energy supply on lunar soil. Specifically, the objective is to establish the architecture of a satellite constellation equipped with solar panels and a laser system capable of transmitting power to a receiver situated on the lunar surface. The primary function of the laser beam within this process is to serve as a conduit for the transmission of energy between distinct locations. By virtue of its unique properties, laser light possesses the capability to traverse through space with negligible dispersion and can be precisely concentrated onto a minute area upon the lunar surface. This characteristic focalization facilitates the concentration of the transmitted energy, resulting in heightened transmission efficiency vis-à-vis alternative energy modalities, notably radio waves. The ultimate outcome of the study will be the analysis of different lunar orbits showing the variation of the power budget depending on the chosen orbit. In addition, key

spacecraft systems such as the laser and EPS will be delineated. Investments in the Artemis program by various government agencies, including NASA [1,2], ESA [3], JAXA, and CSA, have been driven by the desire to explore space, focusing on our solar system, starting from the Moon. One of the program's primary objectives is to establish a permanent lunar base [4], which will serve as a launchpad for the exploration of Mars. The program will also involve studying the psychological and physical response of the astronauts and testing new technologies. The presence of future bases on lunar soil, inhabited by both humans and robots [2], and the increase in the number of lunar missions will result in an increasing need for energy supply on the Moon. Traditional methods of power generation and distribution, based on the installation of photovoltaic panels by sunlight conversion, are not suitable for working in the lunar environment, which has different characteristics to those on Earth. As an example, in [5] a case study of a solar power plant with batteries at the lunar south pole resulted in an estimated area of 282 m$^2$ and a total mass of 68 tons, dropping to 14 tons with the use of fuel cells.

The goal of this work is to find the best architecture that takes advantage of the new idea of power generation by untying the concept of localized generation in the same place where it is used, in this case the lunar soil. The duration of a lunar day, or the time it takes for the Moon to complete one rotation on its axis, is approximately 27.3 Earth days. This means that one lunar day is much longer than a day on Earth. This extended duration of a lunar day has significant implications for energy generation and usage on the Moon. One of the major challenges for energy generation on the Moon is the limited sunlight exposure during the lunar night. Since the Moon rotates slowly, each lunar day is followed by a long lunar night, which lasts for about the same duration. During this night, the areas in shadow experience a complete lack of sunlight. Sunlight is a crucial source of energy for solar power systems, so the absence of sunlight poses a significant challenge. To address this issue, energy systems on the Moon must incorporate energy storage capabilities to store excess energy generated during the lunar day for use during the lunar night. Various energy storage technologies can be employed, such as batteries, fuel cells, or even thermal storage systems. These storage systems help bridge the gap between energy generation and consumption, ensuring a continuous power supply during the lunar night. Efficient energy management and conservation practices are also crucial due to the limited availability of energy resources on the Moon. Since energy cannot be continuously generated during the lunar night, it becomes essential to optimize energy usage and minimize wastage. This requires the adoption of energy-efficient technologies and practices, as well as careful planning and scheduling of energy-intensive activities. In the areas closest to the poles, the first infrastructure for human settlements will be built due to the Moon's slight inclination relative to its axis of rotation and its orbit relative to the ecliptic plane. Furthermore, the Sun is almost constantly above the horizon but doesn't rise more than a few degrees above it in a way that its rays are always almost parallel to the surface. This phenomenon, combined with the morphology of the lunar terrain, allows for areas to be very easily shadowed from the Sun causing further variability in the alternation of light and shadow areas near the poles.

These problems led to the choice of the area near Shackleton Crater [6] as the first exploration area, because its morphology means it is almost constantly in the sunlight with nights no longer than 3–5 days [7,8]. The Shackleton Crater, located near the Moon's south pole, has been a topic of interest for lunar exploration and potential habitation. The selection of this area as a potential exploration site is based on several factors such as water ice deposits, and continuous illumination as aforementioned. Another factor is its scientific value because the permanently shadowed regions within the crater have preserved ancient impact materials and potentially volatile substances in pristine conditions. The last factor is mission safety. Indeed the proximity to the lunar pole enables improved line-of-sight communication with Earth-based mission control and potential communication relays with other lunar outposts.

This paper first includes an analysis of the current solutions proposed by major companies, space agencies and research centers; it then focuses on demonstrating the potential of the new system, considering the choice of the best laser, possible orbits, and a basic sizing of the batteries on the satellite and the receiver on the ground. Finally, a power budget for each scenario is calculated, considering a polar, a frozen and an equatorial orbit with different numbers of satellites and maximum angles of the cone of transmission.

## 2. State of the Art Models

One solution for the problem of lunar energy supply proposed by NASA (in collaboration with Astrobotic Technology of Pittsburgh, Honeybee Robotics, and Lockheed Martin of Littleton) is to install large vertical solar panels (as shown in Figure 1 and discussed in [9]) on the edge of the Shackleton Crater. This site was chosen for its scientific interest and for the amount of available sunlight. This solution would solve the problem of the long lunar night, but would force the installation of solar panels locally in that precise area on the surface. Several companies have proposed and studied alternative solutions to using solar panels, such as nuclear energy exploitation. Rolls Royce has recently secured the first contract with the UK Space Agency [10] to study a nuclear microreactor, as depicted in Figure 2, capable of operating in the lunar environment.

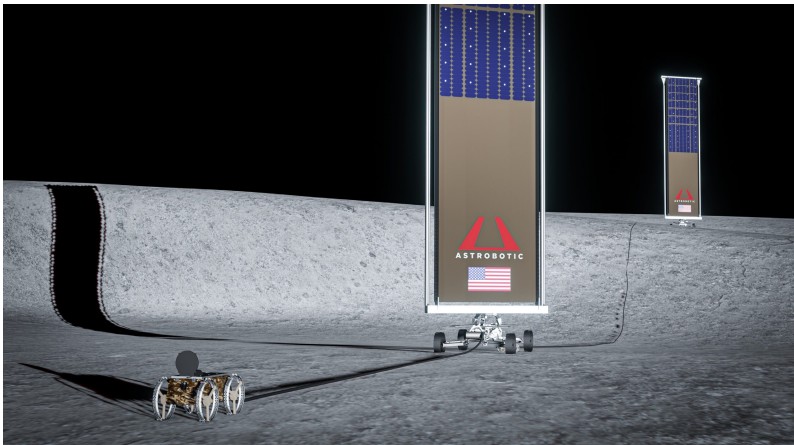

**Figure 1.** Astrobotic vertical panel (Image credits: https://www.astrobotic.com/, accessed on 2 February 2023).

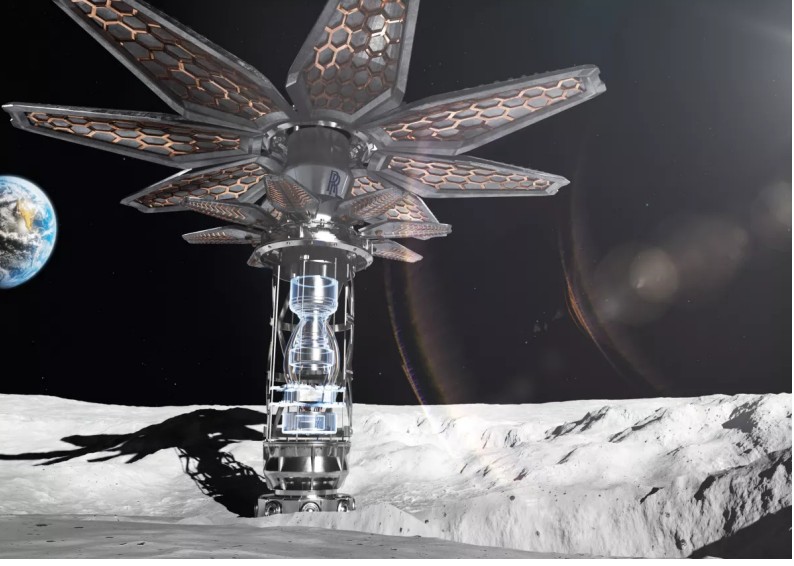

**Figure 2.** Rolls Royce nuclear microreactor (Image credits: Rolls Royce Holdings).

This paper primarily focuses on the topic of wireless power transmission (WPT) in space. The main methods of power transmission over long distances are microwave power transmission (MPT) and laser power transmission (LPT). A major challenge in lunar power systems is the extended duration of lunar nights, which results in the inability of solar panels to generate electricity due to the absence of sunlight. MPT and LPT offer an alternative solution by enabling WPT from a power source located on the Moon's surface or in orbit. This allows for a consistent power supply to lunar habitats, rovers, and other infrastructure even during the prolonged periods of darkness. Another challenge in the lunar environment is the presence of fine dust particles and surface debris that can accumulate on solar panels, reducing their efficiency over time. MPT and LPT can mitigate this issue by eliminating the need for extensive solar arrays on the lunar surface. With WPT, the power generation equipment can be located away from the dusty lunar surface, thereby reducing the impact of dust accumulation and minimizing the need for frequent maintenance. This leads to longer operational lifetimes for power systems. At the lunar poles, sunlight is not uniform because of the Moon's tilt, and some areas are constantly in shadow. MPT and LPT can overcome this challenge by transmitting power from well-illuminated regions to permanently shaded areas. This ensures a continuous power supply to potential habitats, facilities, and scientific installations located in these challenging polar environments. MPT and LPT technologies also offer enhanced efficiency and power density compared to traditional wire-based power transmission systems. They can transmit energy more efficiently over longer distances, reducing energy losses during transmission. This allows for the provision of greater amounts of power to distant lunar sites, meeting the increasing demands for power in future missions and lunar colonies. Furthermore, these technologies are scalable and adaptable, allowing for flexible adjustments to meet the evolving energy needs of the Moon. As lunar missions and settlements grow, MPT and LPT systems can be designed to scale up, providing the necessary flexibility in power distribution and facilitating expansion as needed.

When comparing MPT and LPT as methods to transmit energy to the lunar surface, several factors come into play. When discussing efficiency, it is important to note that LPT systems have the potential to achieve higher levels of efficiency compared to MPT systems. Laser beams can be focused with greater precision, leading to reduced energy losses during transmission. In addition, the MPT systems could have significant path losses depending on the transmission distance. As far as the system costs, MPT systems generally have lower initial costs compared to LPT systems. The infrastructure required for MPT, such as microwave transmitters, receivers, and antennas, is relatively less expensive. However, the costs can vary depending on the scale and complexity of the MPT system. Lastly, in terms of mass, LPT systems typically have a smaller physical footprint and lower mass compared to MPT systems. This is due to the narrow beam of laser transmission, which requires a smaller receiving area. Laser transmitters and receivers can be more compact in size, and the absence of large antennas further contributes to reduced mass requirements.

The historical milestones of WPT using microwaves are further explored in [11]. The paper delves into the details of transmitters and receiving components, including rectennas, a technology that was invented several decades ago [12] but is still under development and has a relatively low technological readiness level (TRL). The basic principle behind MPT is the conversion of electrical energy into microwave energy. This microwave energy is subsequently transmitted to a receiving antenna, where it is reconverted back into electrical energy. The efficiency of this process depends on the frequency of the electromagnetic waves used for the transmission, as well as the distance between the transmitting and receiving antennas. This technology is of great interest in the search for a solution to space-based solar power for the Earth [13], and one of the major challenges in its development is the potential health risks [14] associated with exposure to high levels of microwave radiation.

The topic of LPT is addressed in [15], where it is concluded that "all studies so far essentially concluded that there were no principal technical showstoppers to the concept.

On the other hand, with each redesign cycle based on new technology, the total mass in orbit, cost and complexity of the entire system decreased substantially, indicating a remaining potential for further improvements". The basic principle behind LPT is similar to that of MPT. Electrical energy powers a laser which converts it into electromagnetic energy, which is then transmitted to a receiving station to be converted back into electrical energy. Because laser radiation can be focused into very narrow beams, this technology has the potential to be much more efficient over very long distances compared to other methods of WPT for the same amount of power transmitted. This leads to one of the major challenges for the system, which is the implementation of a very accurate pointing system (with sub-arcsecond accuracy) to allow pointing such a narrow beam over hundreds or thousands of kilometers. The other crucial challenge comes from the need for high-power lasers which generally have an efficiency of at most 50–60%. This means that the satellite must have onboard a highly performant thermal management system to dissipate heat generated by the laser.

In [16] two different WPT systems are compared as methods to send energy to the lunar surface. The location of future research sites on the lunar polar regions has raised the overarching problem of how to supply continuous power to the lunar bases located at the poles. Continuous sustainable power is one of the many challenges in establishing a permanent lunar outpost.

The laser system has an overall lower efficiency, but is worth investigating after comparing the cost and mass of the system, both space- and Moon-segments. An overview of the research in Europe on WPT can be found in [17], where a project from JAXA to create the microwave-based space solar power system (M-SSPS) to send power to Earth is also mentioned. Space-based solar power is the concept of collecting solar power in outer space with satellites and distributing it to Earth. The idea is similar to the one developed in this paper but focuses on the use of microwave radiation for terrestrial applications. The research nevertheless demonstrates how the topic of WPT is fundamental for our future towards the exploitation of more clean energy resources.

## 3. System Architecture and Key Technologies

The solution proposed in this paper is a constellation of satellites orbiting the Moon, which generate energy using solar panels, store it on board, and then transmit the energy through the use of laser beams towards specially designed receivers placed on the lunar surface. With this architecture, it is possible to overcome the limitations associated with lunar day and night cycles, enabling lunar missions and applications that require large amounts of energy. A conceptual diagram of the proposed solution is presented in Figure 3.

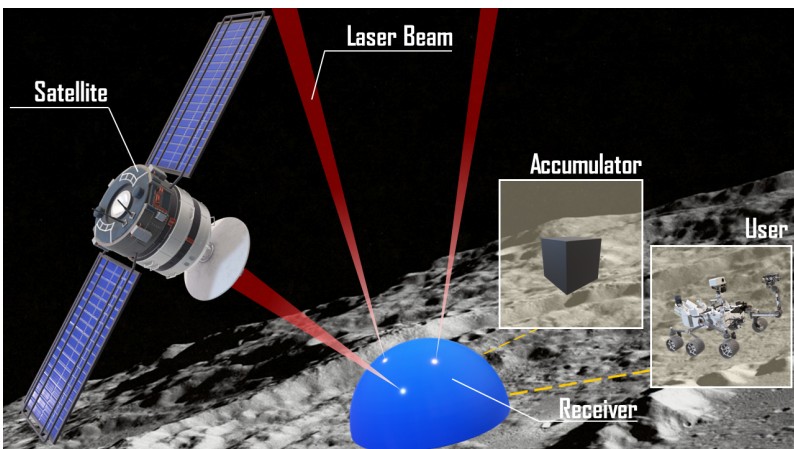

**Figure 3.** Lunar-orbiting satellite constellation solution.

LPT needs key technologies to be enabled, such as a thermal management system, an accurate pointing system, an optical component that collimates the laser beam and

a laser power converter at the receiving end. The first key technology mentioned could include a combination of two different kinds of cooling: forced convection and heat transfer associated with the oscillating heat pipe closed system, as suggested in [18]. To reach accurate pointing, NASA claims in [19] that with a fine guidance sensor (FGS) could obtain a level of stability and precision comparable to being able to hold a laser beam focused on a dime that is 200 miles away (an 18 mm target at a distance from Washington D.C. to New York City). FGSs are very-high-precision sensors used in huge satellites such as the James Webb and the Hubble. Given their extreme precision, they have very significant costs and mass that will therefore need to be carefully evaluated.

### 3.1. Energy Beaming Payload: Fiber Lasers

One of the most important steps towards making these systems more sustainable and attract attention to them is to advance the TRL of the energy-beaming technology (i.e., moving from TRL 4 to 8/9), as explored in depth in [20]. In solid-state lasers, scaling to higher pump powers can lead to a decrease in beam quality and reduced gain due to thermally induced optical aberrations and non-linearities in the gain element, as explored in depth in [21]. Similarly, a larger modal volume can lead to heat removal difficulties in the laser crystal and multi-spatial mode operation. While attempts to optimize the radiance of single gain elements are continuing, many laser systems appear to be reaching their physical limits. Introducing high-brightness laser diodes and double-cladded fibers has made a revolutionary step in fiber laser technology, which has resulted in unprecedented output powers in the CW regime. The proposed architecture as mentioned above makes use of an ytterbium-doped fiber laser, in which the gain medium is an active optical fiber doped with rare elements, as explained in [22]. Fiber laser technology [Figure 4] is rapidly advancing thanks to their advantages over solid-state lasers, and they present characteristics which are critical for WPT. These lasers have been defined the "perfect building blocks" for directed energy applications thanks to their properties of being electrically powered, efficient, modular and reliable with their civilian technology backbone. With the development of new rare-earth-doped fibers such as those with ytterbium and erbium, fiber lasers have the ability to scale to very high powers. The advantages of using a fiber laser include higher beam quality, higher efficiency and greater stability and reliability [23–28]. The pursuit of high-power outputs in industrial lasers and in directed energy weapons has led to an interest in scalable system geometries in which an arbitrary number of identical laser beams can be added together to achieve final output power levels in the order of tens of kilowatts. The use of optical fibers to deliver the output beam allows for greater flexibility in power scaling [29,30]. Although the very narrow output beam has a divergence angle in the order of micro-radians, diffraction over very long distances will cause the beam to spread out as it travels through the vacuum of space. Without proper collimation, the beam would have a final diameter on the Moon's surface of several hundreds of meters, and thus would not be useful for the purpose of power harvesting. In order to reach the surface with a beam of reasonable size, the beam must be expanded at the source. For this purpose, metallic mirrors can be used to create an off-axis telescope, with optical coatings for the VIS and NIR that exhibit extremely low scattered light and absorption losses, both in the order of $10^{-5}$–$10^{-4}$% [31–33].

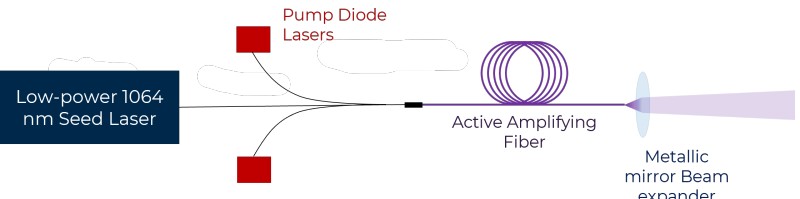

**Figure 4.** High-power fiber laser scheme.

### 3.2. Power Scaling by Laser Beam Combination

There are numerous ways of combining laser beams [Figure 5], and some are better suited to the application than others. In the case of spectral-beam-combining methods, beams of different wavelengths are combined through simple but scalable optical set-ups. The use of diffractive elements (gratings) rather than refractive elements (prisms) is more suited for high-power applications, as they offer more flexibility in design and allow the substrate to be cooled, providing a means to combine closely separated wavelengths while also facilitating thermal control. Another type of system that is important in long-range applications are tiled-aperture systems, in which the beams are co-aligned but are not co-axial. The assumption is that at long range the natural diffraction spreading of the individual beams will ensure that they overlap and form a far-range beam that can be more or less uniform. The tiled approach provides more flexibility than a full aperture system with co-axial beams, and facilitates scalability. It is also not restricted to spectral combining, a factor that is not strictly necessary or required in power transmission systems. The approach to tiled aperture combining systems depends on whether or not the beams are coherent. If the beams are coherent (meaning they require mutual coherence of the combined beams), this means that their relative phase can be controlled such that the wavefronts effectively form a continuous single larger wavefront, and the far-field intensity pattern is set by the diffraction of this single larger wavefront. The consequence is that the intensity of the beam increases as $n^2$, with $n$ being the number of combined beams. For the incoherent case, the far-field intensity pattern is simply the sum of the individual intensity patterns, and the intensity scales only as $n$. The beam combining of laser arrays with high efficiency and good beam quality for power and radiance (brightness) scaling is a long-standing problem in laser technology. Ref. [16] presents a solution to an all-fiber coherent-beam-combining optical system towards a higher output for the fiber laser.

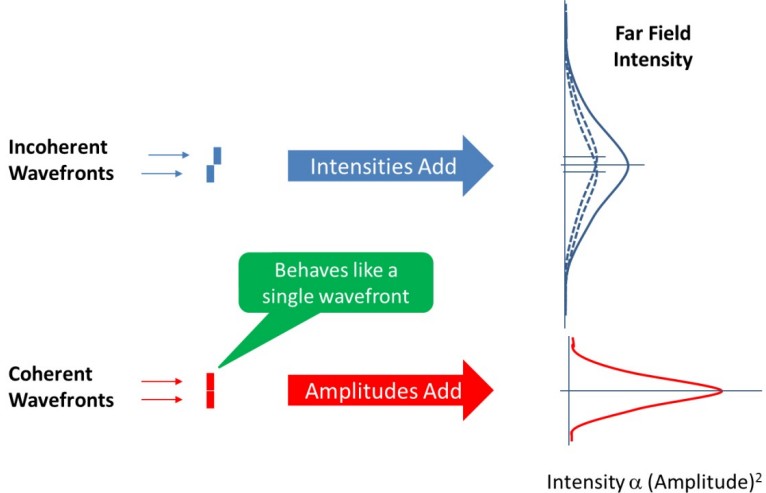

**Figure 5.** Coherent and incoherent combination.

Clearly the advantages of coherent beam combining can be profound, although in reality the wavefront has discontinuities which have the effect of superimposing "ring" structures on the far-field pattern, smearing some of the power into side-lobes of the main beam. This effect may be significant in the application of power transmission over very long distances, in the case that coherent combining methods prove to be suitable for the application. For any tiled-aperture method, it can be demonstrated that there is an optimal spacing between the beams which results in a maximum intensity in the far field, and this will have to be included in future works. To provide the constructive interference that will yield the $n^2$ intensity increase, the beams must be brought into phase alignment, and typically the phase deviations need to be well below 1 rad root mean square. In a laser-beam-combining system on a spacecraft in orbit around the Moon, the stability

requirements for the satellite platform would likely be too strict to be feasible. If this proves to be true, the system would have to be designed to work with an incoherent combination of laser beams that preserves the quality and diffraction limit of the single beams as much as possible. In [34], a practical example of a tiled-aperture high-power coherent beam combiner is given, in which the fibers to be combined are individually positioned to an accuracy of 1 micron. There are some general rules to be considered, but there are likely to be various ways of achieving the final aim, and the optimal solution will depend upon many driving parameters such as cost, performance, power, waveband, etc. For this particular application, new challenges also arise from the environment in which the system has to operate. The harsh radiation and high-temperature gradient that the system would have to endure shorten the list of materials and components that are appropriate for the application. To enlarge the laser beam from its fiber-output diameter to its final diameter suitable for long-distance propagation, the telescope should be designed with mirrors instead of the more commonly used lens, mainly because of the degradation that the latter would undergo when subject to such high radiation.

### 3.3. Receiver Cells and Structure

The receiver being studied is composed of photovoltaic cells specifically designed to convert the laser's monochromatic light, and in this case are also called photonic power converters. The bandgap of their absorber material closely matches the incoming photon energy, allowing higher conversion efficiencies. The cells are arranged on a deployable structure which can be deployed either manually or autonomously on the surface of the Moon. The design of such a cell is presented in [35] and in [36]. Although this type of cell is not currently an off-the-shelf component, like the more widely used solar cells, several European companies and research centers are currently developing this type of technology. Traditional photovoltaic cells are designed to convert a broad range of the solar spectrum and are thus subject to efficiency limitations because of the limited bandgap of the cell materials [37]. This new technology allows to overcome the efficiency limitations of traditional cells by matching the power and wavelength of the laser light with the properties of the photovoltaic cell: in the case of NIR wavelengths, cell structures made of InGaAs material can be used [35,36]. Thanks to the specifics of the materials of which the cell is composed, efficiencies are more than doubled with respect to traditional materials. At the Fraunhofer ISE, a record conversion efficiency of 68.9% was recently achieved with a very thin gallium arsenide photovoltaic cell under monochromatic laser light [38]. As a result, smaller surface areas are used to convert the same amount of power, with lower mass and cost compared to traditional technologies.

### 3.4. Orbital Analysis

Frozen orbits are special orbits [39–43] that allow a satellite to minimize natural drifting due to central body disturbances. This is carried out by carefully choosing orbital parameters, and allows the satellite to maintain its orbital parameters unchanged by rarely requiring orbital corrections. Frozen orbits are particularly useful for long-term missions to the Moon, as they allow satellites to remain in a stable and constant orbit without the need for constant orbital corrections. However, frozen orbits require extreme precision in their planning and maintenance, as even a small deviation from the desired trajectory can cause the satellite to leave the frozen orbit. Several studies have been conducted, particularly on lunar frozen orbits, finding that most low lunar orbits (LLOs) are unstable [44,45]. In 2006 NASA [46] claimed to have only found four frozen orbits, at the following inclinations: 27°, 50°, 76° and 86°. However, every year, different studies demonstrate the existence of new lunar frozen orbits [47]. The studies [48–50] discovered low elliptical lunar frozen orbits with a semi-major axis of about 1800 km.

The determination of the lunar gravity field is based on tracking data from previous Moon missions. In [51], a model of the lunar gravity field using the Lunar Prospector mission from 1998–1999 can be found. Lunar mascons make most low lunar orbits unstable,

and the main disturbance to frozen orbits is studied and modeled in [52]. Frozen orbits have been used in several lunar space missions, including NASA's Lunar Reconnaissance Orbiter (LRO) mission [53] launched in 2009, which studied the lunar surface with advanced scientific instruments and took high-resolution images of the surface. Additionally, frozen orbits have been used for positioning communication satellites, such as the Lunar Atmosphere and Dust Environment Explorer (LADEE), which studied the lunar atmosphere and dust, and the Lunar Reconnaissance Orbiter Communication System (LCRS), which provides communications support for the LRO mission and other future missions to the Moon. Furthermore, ref. [54] presented a new explicit transformation from mean to osculating elements that enables better behavior in terms of long-term stability about the mean values of eccentricity and periapsis, especially for high orbits.

*3.5. On-Board Storage System*

Concerning the storage system, lithium-ion batteries are preferred because they offer high energy density, high efficiency, long service life, low self-discharge, low self-heating rate, and are used in many space applications including interplanetary missions and space stations. In [55], a simulation of battery charge/discharge of a small satellite can be found. This simulation was taken as a starting point for the Simulink model in this paper. The study presented below thus combines an optimization between the number of satellites and the choice of orbit. The main driver in this optimization is to transfer as much energy as possible for the future lunar base from orbit to ground through a high-power laser payload.

## 4. Laser Background Theory

The Gaussian beam model was used to describe the propagation of the laser beam in free space and taken into account for the determination of the power budget. There are many parameters that can be used for the design of a laser beam [56]. Since we are mostly interested in irradiance (power transmitted per unit area) and the size of the beam at the end of propagation, we have taken these variables into account in this work. The irradiance distribution of a 2D Gaussian laser beam $I(r, z)$ is given by the following formula:

$$I(r, z) = I_0 \left( \frac{w_0}{w(z)} \right)^2 \exp \left( \frac{-2r^2}{w(z)^2} \right) \tag{1}$$

where $r$ is the radial distance from the center axis of the beam, $z$ is the axial distance from the beam's waist, $w(z)$ is the radius at which the optical intensity falls to $\frac{1}{e^2}$ of the peak intensity on the plane $z$ and $w_0 = w(0)$ is the beam waist radius. $I_0$ and $w(z)$ are defined as follows:

$$I_0 = \frac{2P_0}{\pi w_0^2} \tag{2}$$

$$w(z) = w_0 \sqrt{1 + \left( \frac{z\lambda}{z_R} \right)^2} \tag{3}$$

where $P_0$ is the total power of the beam and $z_R$ is the Rayleigh range, defined as

$$z_R = \frac{\pi w_0^2 n}{\lambda} \tag{4}$$

where $\lambda$ is the laser's operating wavelength and $n$ is the refractive index of the medium.

## 5. Orbits

In this study, three different types of orbits are considered. The first one is optimal to transmit on a receiver located at the same latitude as the Shackleton Crater, with an inclination of 89.54°. In this case we consider a lunar circular polar orbit with an altitude of 600 km. The satellites are evenly distributed and the true anomaly is computed as follows:

$$\nu_k = \nu_1 + \frac{2\pi(k-1)}{N_{satellites}} \tag{5}$$

where $k$ indicates the $k$-th satellite. This orbit, like all high-altitude lunar orbits, is subject to terrestrial disturbances and so it is astrodynamically inaccurate and would require an oversized propulsive system for station-keeping operations. For the purpose of this study, it is used nevertheless for calculating the limit of transmissible power for a given altitude. In Figure 6, the satellite constellation geometry is shown in this first case. In the following figures, the purple line indicates the orbit while the colored dots indicate the satellites.

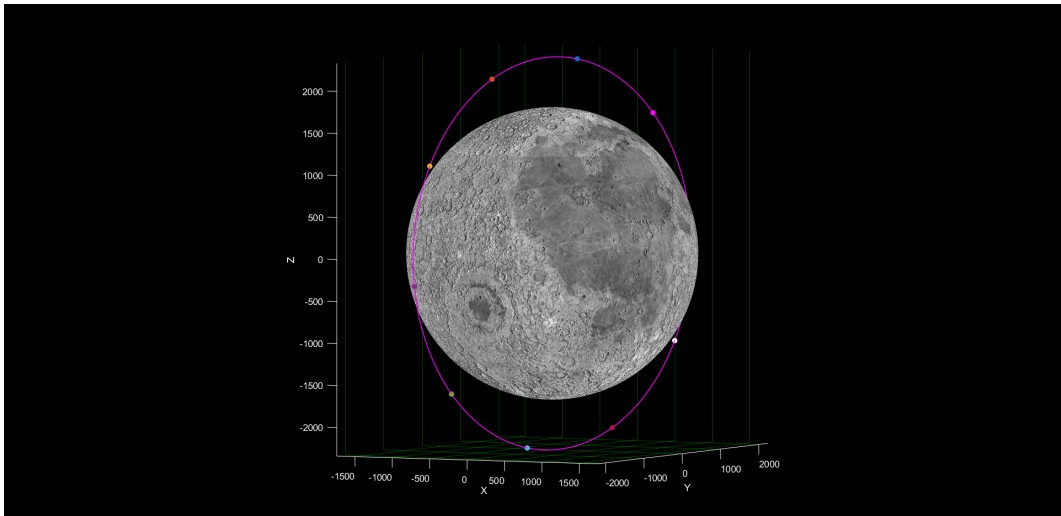

**Figure 6.** Scenario 1: a ten-satellite constellation in lunar circular polar orbit.

The second scenario considers a low elliptical lunar frozen orbit, found in [48]. The orbital parameters are shown in Table 1. The true anomaly is computed as in Equation (5) and the constellation is shown in Figure 7.

**Table 1.** Scenario 2: orbital parameters.

| | |
|---|---|
| $a$ | 1838 [km] |
| $e$ | 0.01 |
| $i$ | 88° |
| $\omega$ | 270° |
| *RAAN* | 0° |

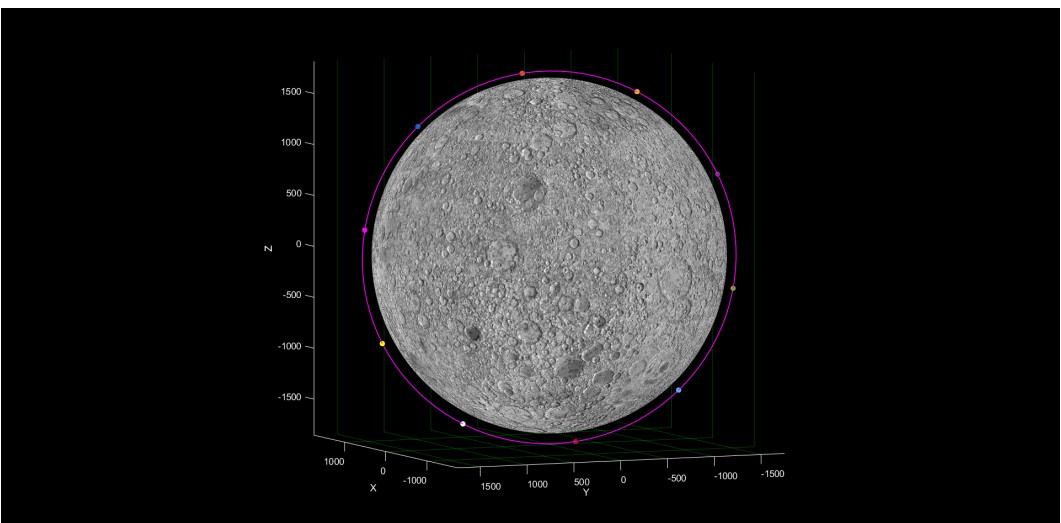

**Figure 7.** Scenario 2: a ten-satellite constellation in low elliptical lunar frozen orbit.

In the last scenario, we assume a receiver on the lunar equator. In this case, an Earth–Moon distant retrograde orbit (DRO) is considered, which is neutrally stable with $A_x = 2538$ km, as defined in Figure 8.

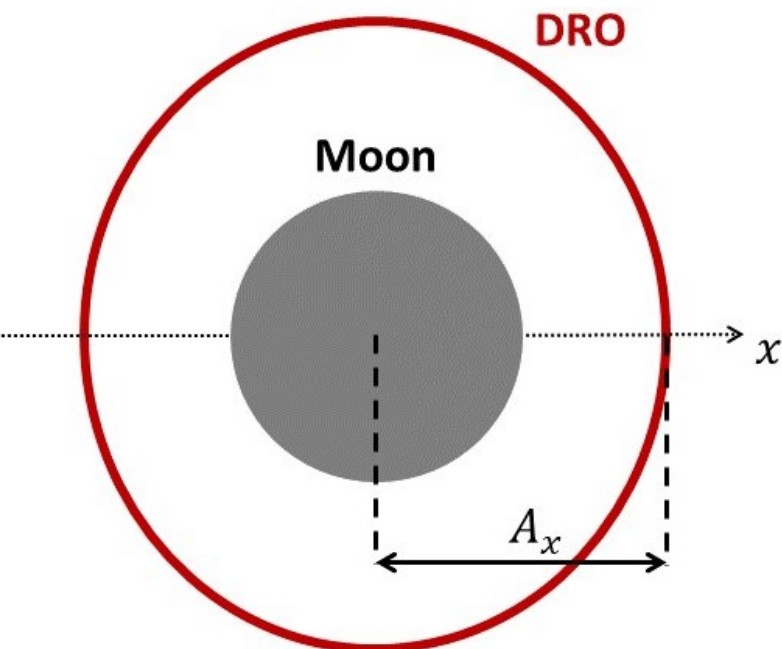

**Figure 8.** DRO schematization.

The circular restricted three body problem (CR3BP) is utilized to model the dynamics of the Earth–Moon system, considering the Earth and Moon as the primary masses. This approach allows for the calculation of periodic orbits in close proximity to the Moon. However, many of these orbits, including Lyapunov and halo orbits, are highly unstable and can come dangerously close to the Moon's surface, sometimes within a few hundred meters. Nonetheless, there exist stable periodic orbits that are sufficiently distant from the Moon's surface, known as Earth–Moon DROs.

Within the CR3BP framework, periodic orbits can be classified into different types based on their symmetry: axis-symmetric, doubly symmetric, and planar (with most DROs falling into the planar and axis-symmetric category). To determine the initial conditions of a desired Earth–Moon DRO, a differential correction method [57] is employed. This method enables the estimation of a set of initial conditions which when integrated using the equations of motion of the CR3BP will result in a periodic orbit over a specified time period, with the final state matching the initial conditions.

Before employing the aforementioned differential correction method, the initial conditions (ICs) for a proposed Earth–Moon DRO are obtained using particle swarm optimization (PSO) [58]. PSO is a heuristic algorithm based on computational swarm intelligence techniques, where a collective group of simple agents, such as schools of fish, bird flocks, or insect colonies, collaboratively solve problems by combining genetic adaptation and social observation to install collective intelligence. The satellite constellation of the third case described above is represented in Figure 9.

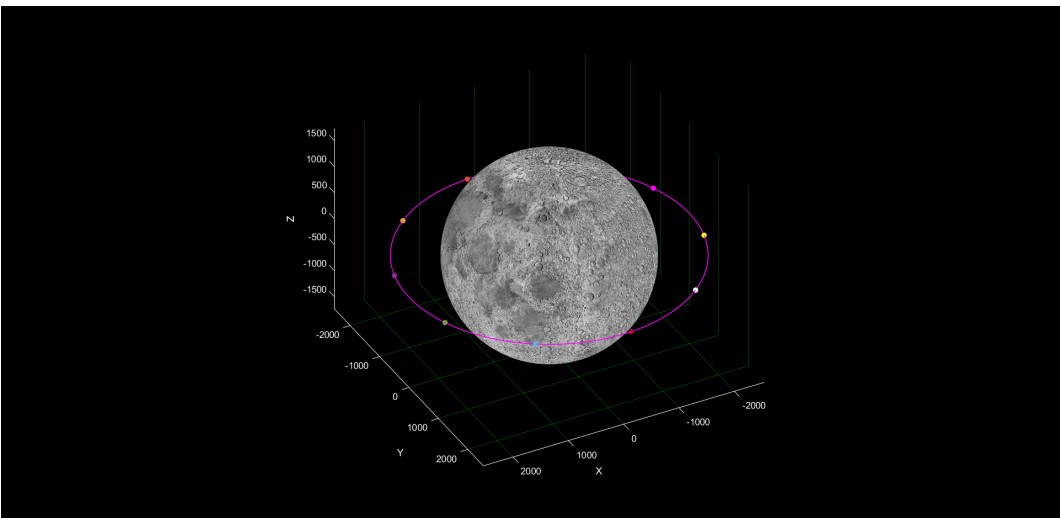

**Figure 9.** Scenario 3: a ten-satellite constellation in Earth–Moon distant retrograde orbit.

## 6. EPS Pre-Design

To simulate multiple charge and discharge cycles of a battery pack, while considering the laser's operating conditions and the satellite orbit, the Simscape Tool from Matlab and Simulink (R2022b) was employed. The sizing scenario investigated pertains to a 600 km circular orbit. In this case, the battery pack was designed by arranging batteries in series and parallel configurations within modules. Subsequently, the battery parameters were determined and remain fixed as follows:

- Cell capacity: 20 Ah.
- Thermal Mass: 400 J/K.
- Cell level coolant thermal path resistance: 1.0 K/W.
- Cell level ambient thermal path resistance: 20 K/W.
- Open-circuit voltage: [3.49–4.19] V.
- Initial State of Charge: 0.3.

In the Simulink model, a constant current and voltage algorithm was implemented to control the charging and discharging of the battery. The battery's discharge, charge, and rest phases are regulated by a function that calculates the satellite's position relative to the transmission cone, whether it is in the light or shadow of the Sun, based on the orbital information.

The electrical power system (EPS) was studied through preliminary sizing, focusing on the battery pack connected for power generation from solar panels and transmission through lasers. The concept involves charging the batteries during the orbital phase outside the receiver's transmission cone and discharging them during laser operation. The charge–discharge cycle was planned to ensure the battery's state of charge remains within the limits of 10% and 90% to preserve its lifespan.

Figure 10 illustrates the behavior of the battery during charging, operating at a constant current of 15 A, and discharging at 50 A. The selected laser has a power output of 30 kW and operates at 600 V. The battery pack consists of two parallel configurations, each comprising two pouches in parallel, with a series connection of 40 modules. Furthermore, two of these assemblies are connected in series, resulting in a total of 320 individual cells.

The battery pack has a weight of 64 kg and occupies a volume of 0.04 m$^3$. The significant weight, considering additional electronic components such as boost converters and switches, is in line with expectations for a satellite where the majority of mass is distributed among lasers and the EPS system.

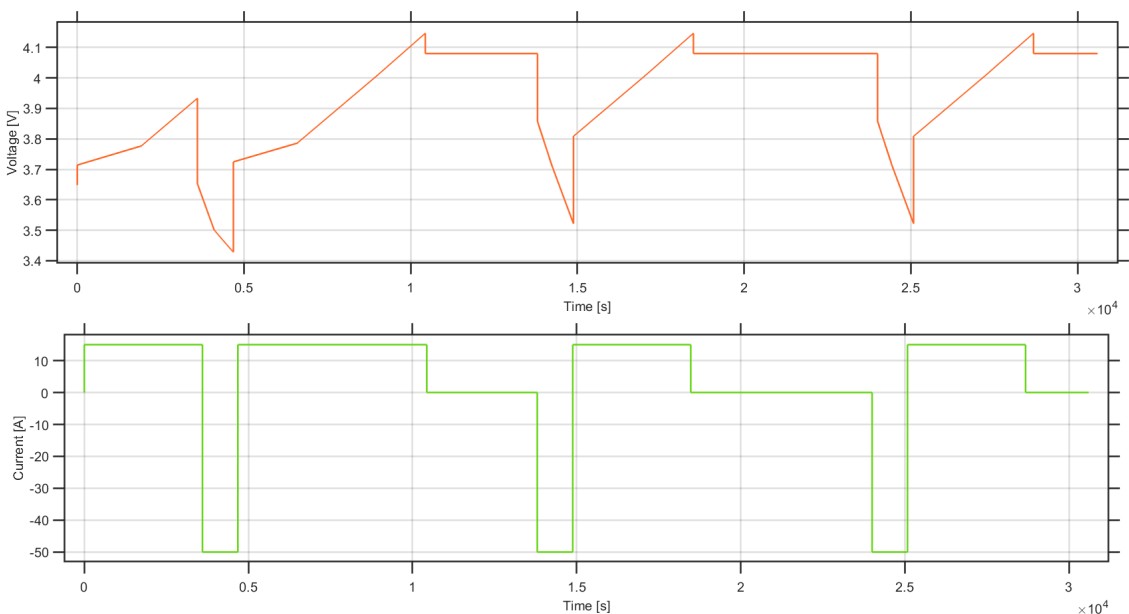

**Figure 10.** Current of the battery pack and voltage of the single cell vs. the time of three orbital periods.

Figure 11 illustrates the optimized state of charge (SoC) of the battery, specifically tailored for the laser system. The battery operation is designed to cease energy storage when the SoC reaches 90% to preserve its operational lifespan. The figure demonstrates the system's capability to recharge during orbital periods that do not involve power transfer. In this study, the system was initially set with a disadvantageous SoC level of 30%, yet it successfully achieves a duty-cycle condition where reliable power transfers can be sustained. The verification process involved combining orbital propagation algorithms while considering shadow periods caused by both the Moon and occasional Earth eclipses. This analysis ensures that the system is capable of consistently performing power transfers despite intermittent shadowing events.

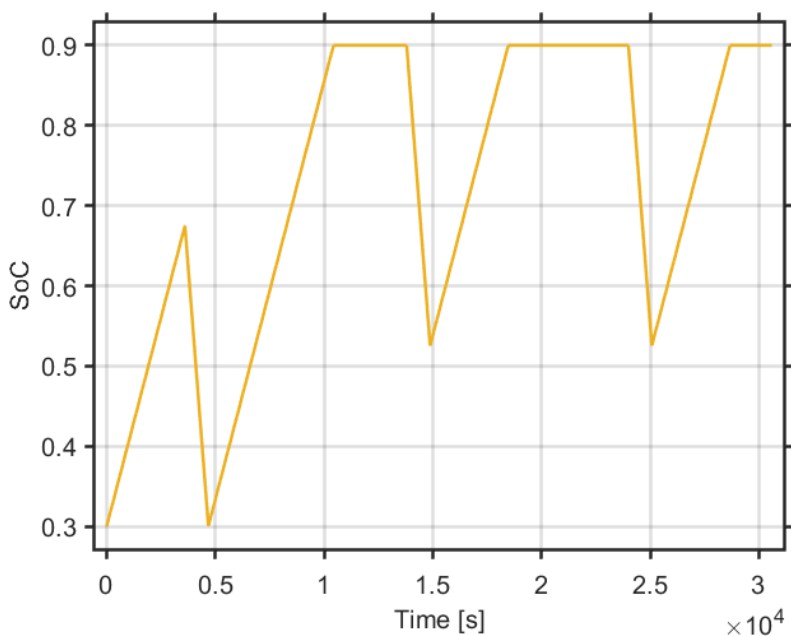

**Figure 11.** SoC of the battery pack vs. the time of three orbital periods.

The estimation of the satellite solar array area takes into account various factors. These include an initial efficiency of 0.34, an inherent degradation factor of 0.77 and an annual

degradation factor of 2.7%. The projected lifetime is 10 years with an efficiency of 0.65 for the paths from the solar panels through the batteries to the individual loads, and an efficiency of 0.85 for the path directly from the arrays to the loads [59]. Based on these considerations, the total computed solar array area is determined to be 22 m$^2$.

## 7. Power Budget

To compute the power provided to a receiver located on the lunar surface, three different scenarios were analyzed, one for each orbit. Every scenario takes into account eight sub-cases, determined by four different angles of transmission and two configurations of the number of satellites in orbit. In fact, in this study we have examined a constellation of three and ten satellites which are evenly distributed around the Moon as explained in Section 5. We chose a constellation of three and ten satellites to compare the difference between a small constellation (lower economic cost) and a large constellation (higher economic cost) in terms of the energy delivered. In two out of the three scenarios the receiver is located at the south pole at a latitude of 89.54° S (the same as the Shackleton Crater), while the last one considers a receiver placed on the lunar equator. As mentioned, each of these cases was then studied for different angles of the cone of transmission: 50°, 60°, 70° and 80°. When the satellite enters into the cone shown in Figure 12, the WPT begins. In this study we computed the irradiance that gives a receiver with the Gaussian beam model explained in Section 4. The laser data used for the simulation are as follows:

- Laser wavelength: $\lambda = 1064$ nm;
- Laser peak power: $P_L = 30$ kW;
- Laser electrical efficiency: $\eta_L = 0.5$;
- Laser waist radius: $w_0 = 250$ mm.

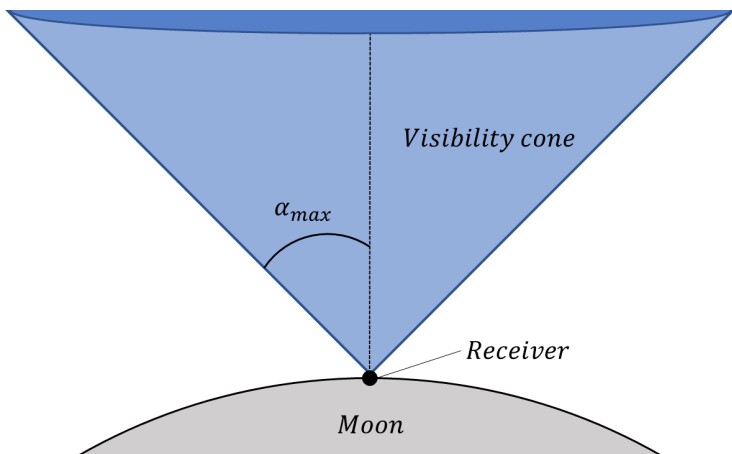

**Figure 12.** Transmission cone schematization.

For each position of the satellite the code computes the distance between the satellite and the receiver $z$. If the satellite is inside the previous transmission cone set, the code computes the irradiance $I_r$ for each cell in the receiver. Therefore, the power received from the receiver $P_r$ at time $t$ becomes

$$P_{r_{(t)}} = \sum_{k=1}^{N_{cells}} I_{r_{(t,k)}} \cdot A_{(k)} \cdot \cos\theta_{(t,k)} \cdot \eta_{cell} \tag{6}$$

where $\eta_{cell}$ is the cell efficiency considered equal to 0.689, the subscript $r$ means received, while the subscript $k$ indicates the $k$-th cell.

In the graphs to follow, the red lines represent the energy transmitted from the laser to the receiver as the satellite enters the cone of transmission. The dashed black line, on the other hand, represents the average power that that architecture manages to send to the Moon.

### 7.1. Scenario 1

Considering a lunar circular polar orbit at 600 km, a change in the maximum transmission angle greatly affects the power the system is capable of transmitting, as shown in Table 2. With three satellites at 600 km, it is possible to transmit on average until 5.68 kW of power, which correspond to 136 kWh in one day. This power would not be enough to completely power a lunar base but would certainly be a great resource for lunar night and rover use. To increase this power, the trade-off consists of the number of satellites, the orbit of the constellation and the power of the laser and its efficiency. In particular, the laser power could potentially reach hundreds of kW, but it would greatly increase the weight and complexity of the thermal dissipation system. With 10 satellites, the energy transmitted increases considerably. In the best case, the constellation would be able to energetically help a possible lunar base for almost 20% of its total needs (set to 2.4 MWh). Indeed the ten-satellite constellation could provide 454 kWh of energy to a lunar base in one day. In Figure 13, we can note that the base load that a single satellite can transmit is equal to 10.34 kW, which exactly corresponds to the following calculation

$$P_r = P_L \cdot \eta_L \cdot \eta_{cell} \tag{7}$$

In this way, when two satellites are at the same time into the visibility cone, the power peak increase to 20.68 kW as shown in Figure 14.

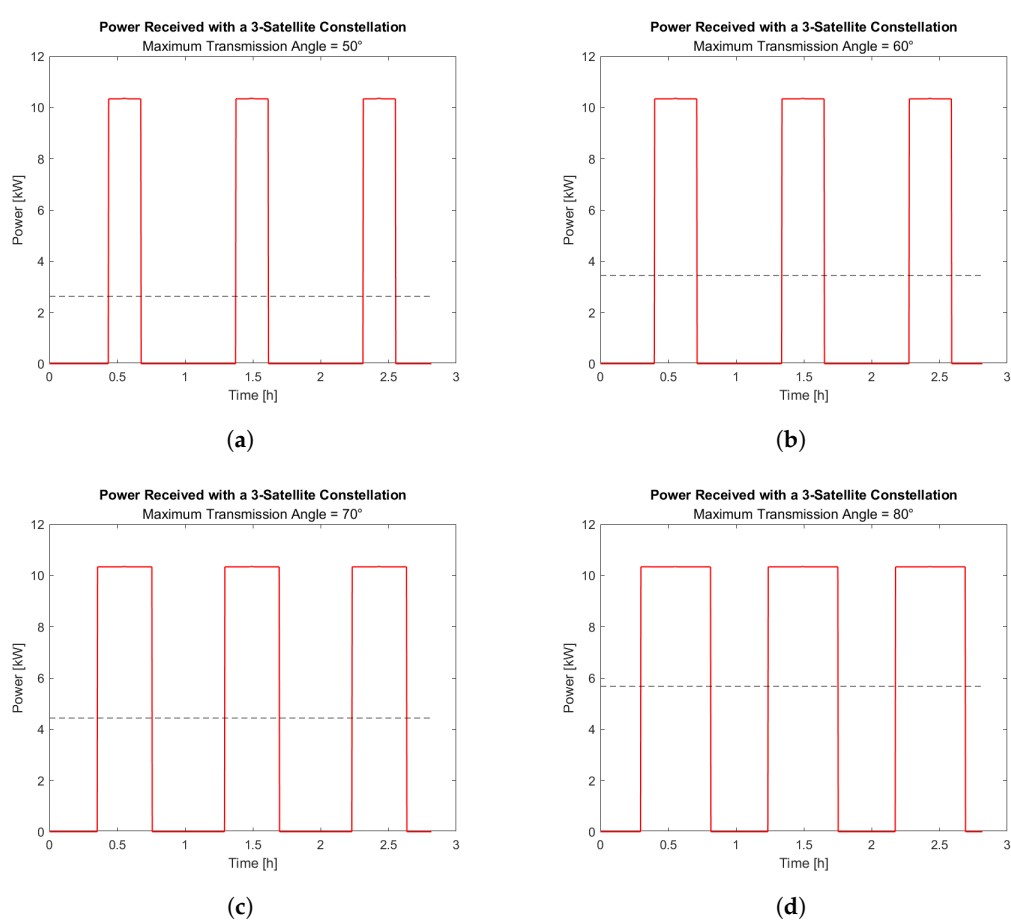

**Figure 13.** Power vs. time with a three-satellite constellation placed in lunar circular polar orbit with different maximum transmission angles. (**a**) $\alpha_{max} = 50°$; (**b**) $\alpha_{max} = 60°$; (**c**) $\alpha_{max} = 70°$; (**d**) $\alpha_{max} = 80°$.

**Table 2.** Power converted in scenario 1.

| | \multicolumn{8}{c}{**Scenario 1: Lunar Circular Polar Orbit**} |
|---|---|---|---|---|---|---|---|---|
| $\alpha_{max}$ | \multicolumn{2}{c}{50°} | \multicolumn{2}{c}{60°} | \multicolumn{2}{c}{70°} | \multicolumn{2}{c}{80°} |
| # sat | 3 | 10 | 3 | 10 | 3 | 10 | 3 | 10 |
| $P$ (kW) | 2.64 | 8.78 | 3.44 | 11.46 | 4.44 | 14.76 | 5.68 | 18.92 |

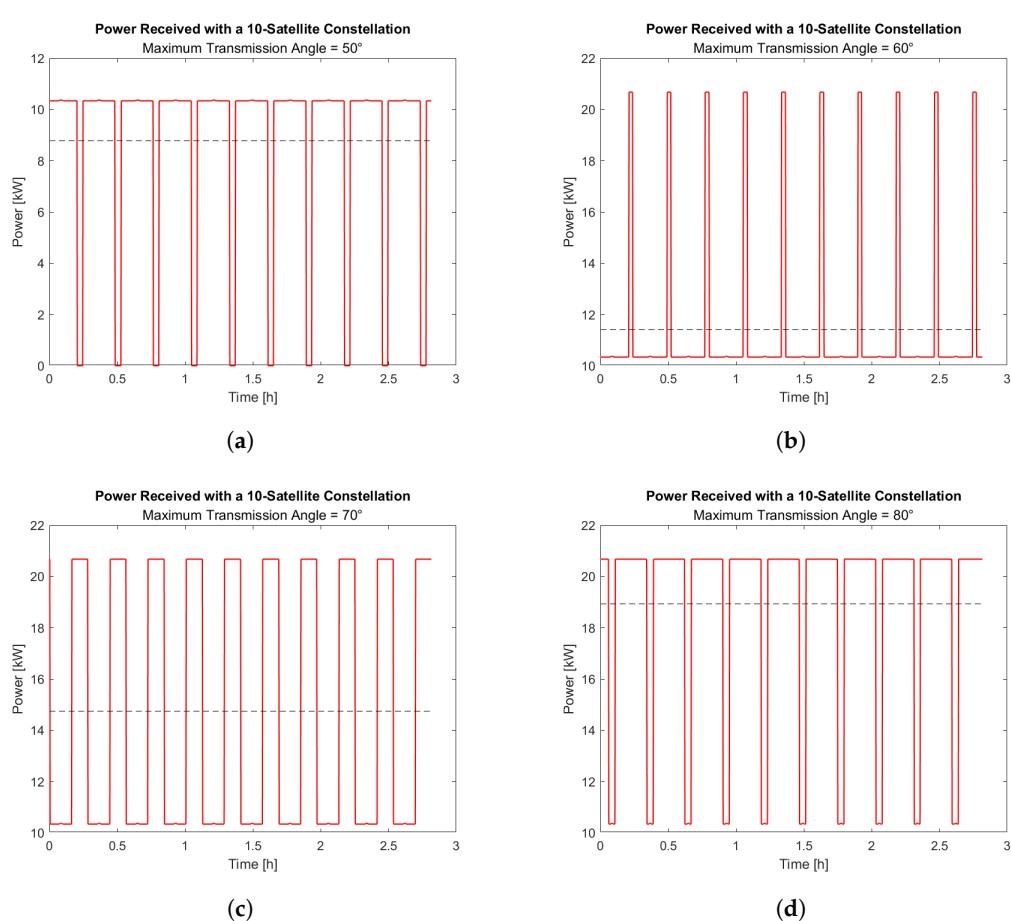

**Figure 14.** Power vs. time with a 10-satellite constellation placed in lunar circular polar orbit with different maximum transmission angles. (**a**) $\alpha_{max} = 50°$; (**b**) $\alpha_{max} = 60°$; (**c**) $\alpha_{max} = 70°$; (**d**) $\alpha_{max} = 80°$.

### 7.2. Scenario 2

In this scenario, the satellites orbit in a low elliptical lunar frozen orbit with a mean altitude equal to 100 km. As the altitude is much lower than in the first scenario, the satellite is faster and the arc of the cone is smaller. For this reason the time spent by the satellite in the transmission cone is much lower than in the first scenario as shown in Figure 15, and consequently the average power transmitted is lower. We can note in Table 3 that, as in the first scenario, increasing $\alpha_{max}$ increases the power transmitted. The sub-case of a 10-satellite constellation shown in Figure 16 shows how there is no transmission overlap between two satellites; consequently, in this scenario the power peaks higher than the base load of the single satellite (10.34 kW) are not obtained compared to the other two scenarios. Concluding in the second scenario, the best three-satellite constellation case could provide 46 kWh of energy in one day, while the best 10-satellite constellation case could provide 155 kWh of energy in one day.

**Table 3.** Power converted in scenario 2.

| | Scenario 2: Low Elliptical Lunar Frozen Orbit | | | | | | | |
|---|---|---|---|---|---|---|---|---|
| $\alpha_{max}$ | 50° | | 60° | | 70° | | 80° | |
| # sat | 3 | 10 | 3 | 10 | 3 | 10 | 3 | 10 |
| $P$ (kW) | 0.56 | 2.11 | 0.81 | 2.89 | 1.21 | 4.17 | 1.92 | 6.46 |

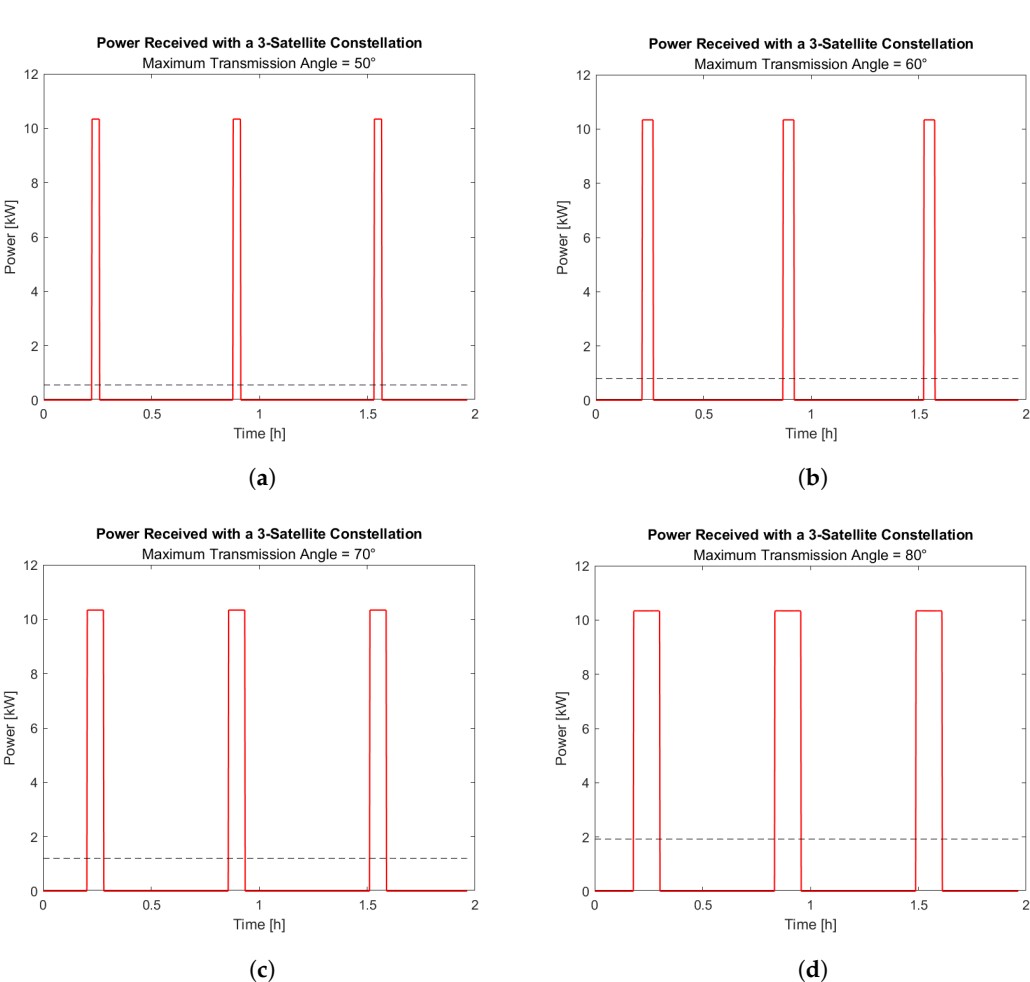

**Figure 15.** Power vs. time with a three-satellite constellation placed in low elliptical lunar frozen orbit with different maximum transmission angles. (**a**) $\alpha_{max} = 50°$; (**b**) $\alpha_{max} = 60°$; (**c**) $\alpha_{max} = 70°$; (**d**) $\alpha_{max} = 80°$.

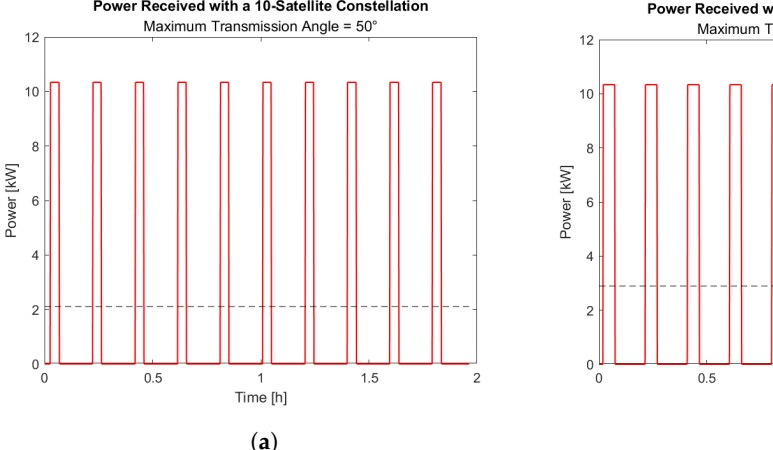

**Figure 16.** *Cont.*

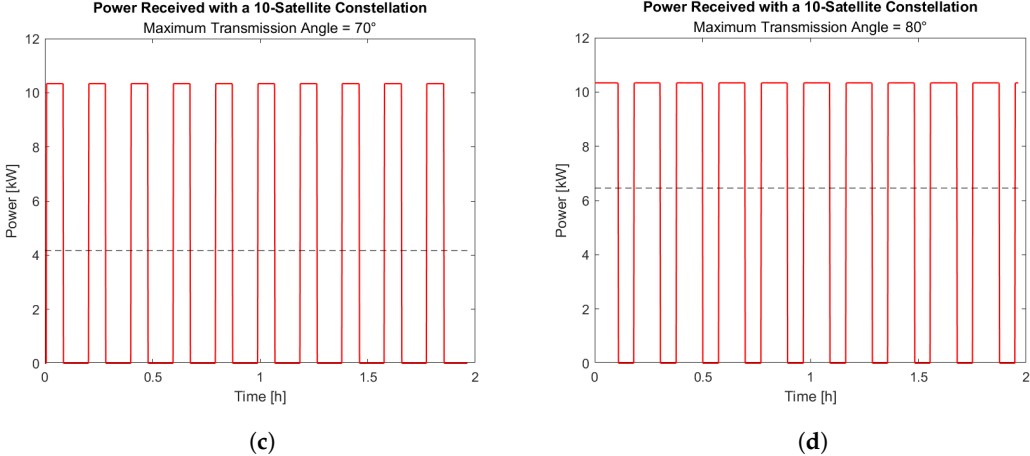

**(c)**                                    **(d)**

**Figure 16.** Power vs. time with a 10-satellite constellation placed in low elliptical lunar frozen orbit with different maximum transmission angles. (**a**) $\alpha_{max} = 50°$; (**b**) $\alpha_{max} = 60°$; (**c**) $\alpha_{max} = 70°$; (**d**) $\alpha_{max} = 80°$.

### 7.3. Scenario 3

In the last scenario considered, where satellites orbit in an Earth–Moon DRO, the power transmitted is higher than the other two scenarios as shown in Table 4 because the average altitude is about 800 km. Figure 17 makes it possible to appreciate how the transmission time increases. In the 10-satellite constellation case shown in Figure 18, it is possible to note the aforementioned power peaks of 20.68 kW because of the simultaneous transmission of two satellites. In the sub-case with $\alpha_{max} = 80°$, we can figure out how three satellites transmit together for a few minutes to obtain power peaks of three times the base load (31.02 kW). Concluding in the third scenario, the best three-satellite constellation case could provide 155 kWh of energy in one day, while the best 10-satellite constellation case could provide 519 kWh of energy in one day.

**Table 4.** Power converted in scenario 3.

| Scenario 3: Earth-Moon Distant Retrograde Orbit | | | | | | | |
|---|---|---|---|---|---|---|---|
| $\alpha_{max}$ | 50° | | 60° | | 70° | | 80° |
| # sat | 3 | 10 | 3 | 10 | 3 | 10 | 3 | 10 |
| $P$ (kW) | 3.13 | 10.56 | 4.05 | 13.55 | 5.15 | 17.20 | 6.46 | 21.61 |

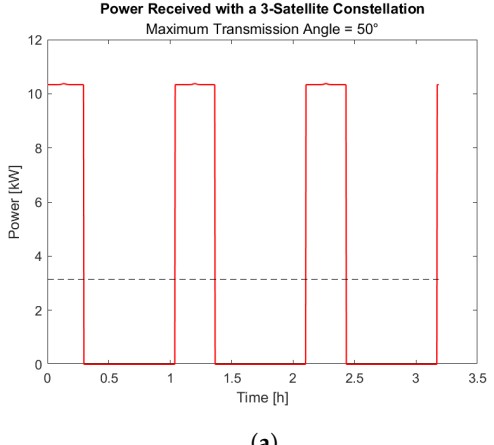
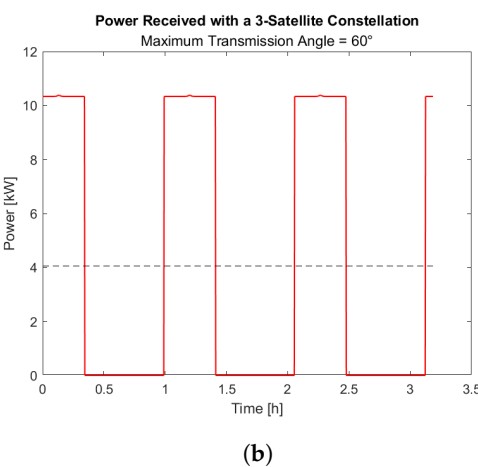

**(a)**                                    **(b)**

**Figure 17.** *Cont.*

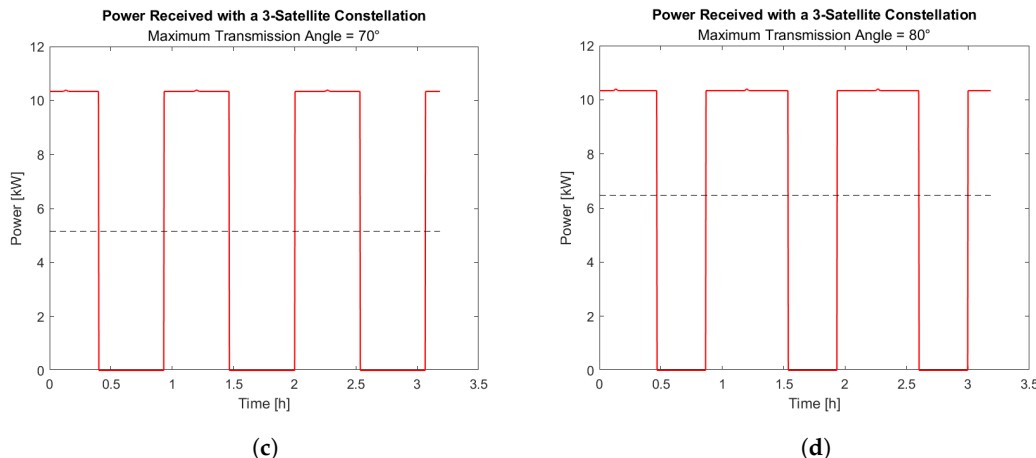

**Figure 17.** Power vs. time with a three-satellite constellation placed in an Earth–Moon DRO with different maximum transmission angles. (**a**) $\alpha_{max} = 50°$; (**b**) $\alpha_{max} = 60°$; (**c**) $\alpha_{max} = 70°$; (**d**) $\alpha_{max} = 80°$.

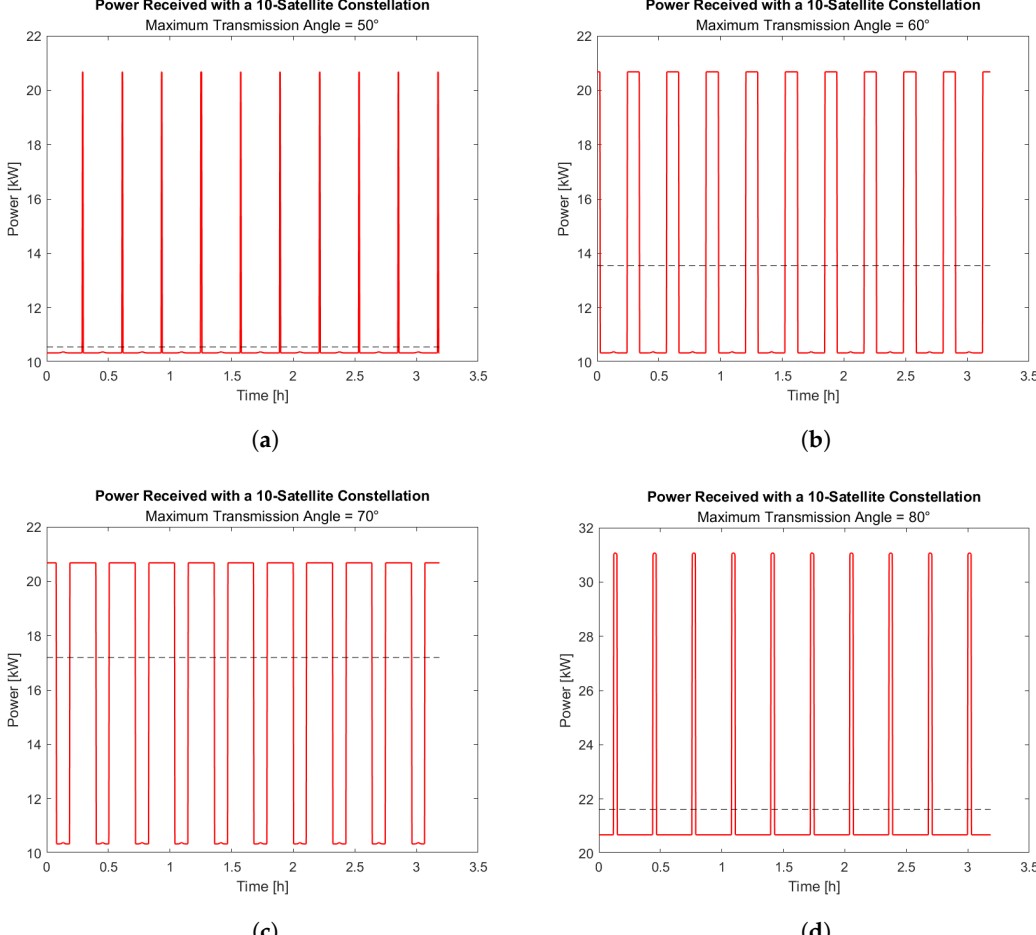

**Figure 18.** Power vs. time with a 10-satellite constellation placed in an Earth–Moon DRO with different maximum transmission angles. (**a**) $\alpha_{max} = 50°$; (**b**) $\alpha_{max} = 60°$; (**c**) $\alpha_{max} = 70°$; (**d**) $\alpha_{max} = 80°$.

## 8. Discussion and Conclusions

This paper aimed to demonstrate the potential of the proposed system focusing on the selection of the optimal laser, different orbits, and initial sizing of satellite batteries and ground receivers. Additionally, a power budget analysis was conducted for various scenarios considering different orbits, Moon bases, and system architectures. With the renewed interest in lunar exploration and the increasing number of planned lunar missions, the energy demand is expected to rise. The highlighted system offers scalability in meeting

the power requirements without the need for designing and launching new technologies for each specific mission, unlike vertical solar panels or future lunar nuclear reactors.

The obtained results indicate that a constellation of 10 satellites could potentially provide up to 519 kWh of power in a single day, under the best sub-case scenario, utilizing current technologies. Further advancements in laser technology will increase the available power onboard and enable new possibilities for WPT in future lunar exploration. However, the amount of power transmitted is closely tied to the duration of time a satellite remains within the transmission cone. Future developments in pointing system accuracy may allow for satellites to be placed in higher orbits, resulting in longer durations within the cone and higher power transmission capabilities. The adaptability of satellite orbital disposal to the needs of the lunar base is a significant advantage of this solution. For instance, in scenarios requiring power peaks for short durations, such as mining activities, satellites can be positioned in close proximity to each other (with slight differences in true anomalies), enabling power peaks equivalent to $N \times 10.34$ kW, where $N$ represents the number of satellites within the transmission cone. The choice of laser technology demonstrates its suitability for transmitting specific power values, although further development and TRL advancements are necessary. Future focus on the development of laser beam sources will play a crucial role in improving the TRL of the system, setting the stage for their use in upcoming space missions.

The overall efficiency of the system primarily depends on the laser efficiency and receiver cell efficiency. Starting from solar power, considerations include the solar panel efficiency of the satellite and distribution losses from the battery to the laser. Other losses related to the shape and size of the receiver can be minimized by utilizing the proposed design.

Future studies will focus on identifying unexplored orbits that facilitate power transmission to key surface points of interest, such as the lunar base's south pole. These investigations will aim to minimize fuel requirements for maintaining stability. Furthermore, the feasibility of long-duration transmissions, such as in the $\alpha_{max} = 80°$ case, will be examined, assessing the limits of the satellite's thermal management system and pointing system, allowing for transmission over greater distances. The development of these technologies could significantly increase the amount of energy transmitted.

In general, the future of this field is expected to witness the advancement of high-power lasers for increasingly effective space applications, leading to substantial improvements in the amount of energy transmitted. Once the architecture has been validated in all aspects, thanks to the system's scalability, the transmitted energy can be increased by expanding the size of the satellite constellation, as explained earlier.

Moreover, one of the strengths of this mode of power generation and transmission is that it can potentially act on the entire lunar surface. This system scalability can have a significant impact for many planned missions and not just for the lunar base energy supply at the lunar south pole. This application may be crucial for the many missions that aim to explore lunar regions never reached by sunlight, and therefore this is interesting for the potential presence of frozen water. Likewise, this may be used for the exploration of natural sites such as lavatubes, caves that in the future may offer natural shelter for establishing new human settlements.

**Author Contributions:** Conceptualization, F.L., A.M., G.M., D.E.S. and A.V.; methodology, F.L.; software, F.L., D.E.S. and A.V.; validation, F.L. and D.E.S.; formal analysis, F.L.; investigation, A.M., G.M. and A.V.; data curation, F.L. and D.E.S.; writing—original draft preparation, F.L., A.M. and D.E.S.; writing—review and editing, G.M. and A.V.; visualization, F.L.; supervision, S.M.; project administration, S.M. All authors have read and agreed to the published version of the manuscript.

**Funding:** The authors would like to thank Fondazione Compagnia di San Paolo with the support of the venture capital holding subsidiary LIFTT SpA, for supporting this research under the program: 'Convezione CSP-POLITO-PoC Instrument Linea 1–PoC Launchpad-Metodo e sistema di trasmissione wireless di energia nello spazio' SIME 2022.2239.

**Data Availability Statement:** All the associated data is contained within the article.

**Conflicts of Interest:** The authors declare no conflict of interest.

## Abbreviations

The following abbreviations are used in this manuscript:

| | |
|---|---|
| NASA | National Aeronautics and Space Administration |
| ESA | European Space Agency |
| JAXA | Japan Aerospace Exploration Agency |
| CSA | Canadian Space Agency |
| MPT | Microwave power transmission |
| LPT | Laser power transmission |
| TRL | Technology readiness level |
| WPT | Wireless power transmission |
| M-SSPS | Microwave-based space solar power system |
| FGS | Fine guidance sensor |
| VIS | Visible |
| NIR | Near-infrared |
| r.m.s | Root mean square |
| InGaAs | Indium gallium arsenide |
| ISE | Institute for Solar Energy |
| LLOs | Low lunar orbits |
| LRO | Lunar Reconnaissance orbiter |
| LADEE | Lunar Atmosphere and Dust Environment Explorer |
| LCRS | Lunar Reconnaissance Orbiter Communication System |
| CR3BP | Circular restricted three body problem |
| ICs | Initial conditions |
| PSO | Particle swarm optimization |
| EPS | Electrical power system |
| SoC | State of charge |
| DRO | Distant retrograde orbit |
| RAAN | Right ascension of the ascending node |

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
