# Peer review of "A Lunar-Orbiting Satellite Constellation for Wireless Energy Supply"

_aerospace, doi:10.3390/aerospace10110919_

Round 1
Reviewer 1 Report
Comments and Suggestions for Authors
LUNAR-ORBITING SATELLITE CONSTELLATION FOR
WIRELESS ENERGY SUPPLY
Dear authors,
The study defined a lunar orbit system that provides power to the lunar surface through wireless power transmission using a constellation of satellites. The authors evaluated possible configurations to satisfy the required power for a lunar base, estimated at approximately 100 kW. Three different orbits were analyzed, one polar, one frozen and one equatorial with different numbers of satellites and different angles of the receiver’s cone of transmission.
In general, the work is well presented, reasoned and presents good results. The results may contribute to future installations on lunar soil. I consider that the work has structure and results to be published.
Here are just a few minor suggestions and/or corrections.
1- Line 33: In “282 m2” , please replace by 282 m2
2- Line 77: In “with low TRL” with low please replace by Technological Readiness Level (TRL). Because this was only defined on line 131.
3- Line 233: In “Frozen orbits are special orbits” I suggest “Frozen orbits are special orbits ([I,II,III,IV,V,VI,49])”
[I] Ely, T.A., Lieb, E.: Constellations of elliptical inclined lunar orbits providing polar and global coverage. J. Astronaut. Sci. 54(1), 53–67 (2006)
[II] Folta, D., Quinn, D.: Lunar Frozen Orbits, Paper AIAA 2006-6749, Aug (2006)
[III] Carvalho, J. P. S., Vilhena de Moraes, R. & Prado, A.F.B.A. Some orbital characteristics of lunar artificial satellites. Celest Mech Dyn Astr 108, 371–388 (2010). https://doi.org/10.1007/s10569-010-9310-6
[IV] Jean Paulo dos Santos Carvalho, Rodolpho Vilhena de Moraes, Antônio Fernando Bertachini de Almeida Prado, "Planetary Satellite Orbiters: Applications for the Moon", Mathematical Problems in Engineering, vol. 2011, Article ID 187478, 19 pages, 2011. https://doi.org/10.1155/2011/187478
[V] Abad, A., Elipe, A., Tresaco, E.: Analytical model to find frozen orbits for a lunar orbiter. J. Guid. Control Dyn. 32(3), 888–898 (2009)
[VI] Elipe, A., Lara, M.: Frozen orbits about Moon. J. Guid. Control Dyn. 26(2), 238–243 (2003)
4- Line 240: In “Several studies (Please cite some references here.) have been conducted particularly on Lunar Frozen Orbits, finding that most Low Lunar Orbits (LLOs) are unstable.”
5- Line 300: In “ km” , please replace by km. See also line 333 (“m”). I suggest using measurement units without italics
6- The legend of some figures are too small.
7- In “In fact, in this study we have examined a constellation of 3 and 10 satellites which are evenly distributed around the moon”
8- Why 3 and 10 satellites? I think I should mention this in the text.
9- The formatting of the list of references should be revised. Because some references are not in the correct model. For example,
See references [3,4,5], [7], [41], [44], [47], among other references
In “Lara, M.; Saedeleer, B.D.; Ferrer, S. PRELIMINARY DESIGN OF LOW LUNAR ORBITS. 2009.” please replace by
Lara, M., De Saedeleer, B., Ferrer, S. “Preliminary design of low lunar orbits,” in Proceedings of the 21st International Symposium on Space Flight Dynamics, pp. 1–15, Toulouse, France, 2009.
[47]- Folta, D., Quinn, D.: Lunar Frozen Orbits, Paper AIAA 2006-6749, Aug (2006)

Author Response
Dear Reviewer,
We sincerely appreciate the thorough attention and review given to our work. We carefully considered every comment and used them to improve our research.
We sincerely appreciate the valuable feedback provided, which helped us enhance the quality of our work.
Please see the attachment.
Kind regards.

Reviewer 2 Report
Comments and Suggestions for Authors
This paper presents a lunar orbiting satellite constellation system that utilizes wireless power transmission to provide energy to the lunar surface, overcoming limitations associated with lunar day and night cycles. The system's architecture involves a constellation of satellites with solar arrays and batteries that transmit power through lasers to specially designed receivers on the lunar surface, offering scalability and potential advancements in laser technology for future lunar missions. These comments may help authors to improve their paper:
1- Abstract- Consider providing a clear statement of the research objective or hypothesis to give readers a better understanding of the paper's focus.
2- Begin the introduction by providing a clear and concise statement of the research problem or objective. What specific aspect of energy supply on the Moon is being addressed in this paper?
3- When introducing the concept of lunar rotation and the resulting duration of a lunar day, consider explaining the implications of this on energy generation and usage. How does the limited sunlight exposure during the lunar night impact energy availability and storage requirements?
4- In the discussion of the areas near the poles and the choice of the Shackleton Crater, clarify why this area was chosen as the first exploration site.
5- Consider expanding on the significance and potential impact of the proposed system.
6- Specify the scope and objectives of the analysis conducted in the paper more clearly in the introduction section.
7- How do microwave power transmission (MPT) and laser power transmission (LPT) address the challenges unique to the lunar environment?
8- When comparing MPT and LPT as methods to send energy to the lunar soil, provide a more detailed analysis of their relative efficiencies, costs, and masses.
9- Clarify the role of the laser beams in transmitting the energy from the satellites to the lunar surface.
10- How does the FGS contribute to achieving accurate laser pointing? Are there any limitations or challenges associated with its use?
11- When discussing the choice of laser technology, consider addressing the Technology Readiness Level (TRL) and any necessary advancements or research required to increase its TRL.
12- Consider providing more information on the overall efficiency and feasibility of the proposed system.
13- What specific developments or improvements related to this research can be expected in the future, and how will they enhance the system's performance?
Comments on the Quality of English LanguageThere are a few areas where the language could be improved for better clarity and flow. Some sentences are quite long and complex, which could be broken down into smaller sentences for easier understanding. Additionally, there are a few instances of repetition and awkward phrasing that could be refined. Overall, with some minor editing and improvements, the English writing of the paper can be further enhanced.
Author Response
Dear Reviewer,
We sincerely appreciate the thorough attention and review given to our work. We carefully considered every comment and used them to improve our research. Below is a list of all the proposed revisions, along with a brief description of the modifications we made.
We sincerely appreciate the valuable feedback provided, which helped us enhance the quality of our work.
Please see the attachment.
Kind regards.
1- Abstract- Consider providing a clear statement of the research objective or hypothesis to give readers a better understanding of the paper's focus.
A clear and concise sentence was added about the objective of the paper and the subsystems it focuses on (rows 6-8).
2- Begin the introduction by providing a clear and concise statement of the research problem or objective. What specific aspect of energy supply on the Moon is being addressed in this paper?
An opening paragraph was added explaining the focus of the paper and the central role of the laser system (rows 26-41).
3- When introducing the concept of lunar rotation and the resulting duration of a lunar day, consider explaining the implications of this on energy generation and usage. How does the limited sunlight exposure during the lunar night impact energy availability and storage requirements?
The concept was better explained in the introduction (rows 59-75).
4- In the discussion of the areas near the poles and the choice of the Shackleton Crater, clarify why this area was chosen as the first exploration site.
Explained the various factors that led to the choice in the introduction (rows 85-93).
5- Consider expanding on the significance and potential impact of the proposed system.
Explained better at the end of the conclusion (rows 607-614).
6- Specify the scope and objectives of the analysis conducted in the paper more clearly in the introduction section.
Added at the beginning of the introduction (rows 17-22).
7- How do microwave power transmission (MPT) and laser power transmission (LPT) address the challenges unique to the lunar environment?
A paragraph was written comparing these two technologies in various aspects (rows 113-152).
8- When comparing MPT and LPT as methods to send energy to the lunar soil, provide a more detailed analysis of their relative efficiencies, costs, and masses.
See point 7.
9- Clarify the role of the laser beams in transmitting the energy from the satellites to the lunar surface.
Added to the introduction when talking about power transmission (rows 29-35).
10- How does the FGS contribute to achieving accurate laser pointing? Are there any limitations or challenges associated with its use?
The FGS are only hinted at to give an idea of the complexity of the control system, which may not be without such a heavy and complex sensor. A sentence was added to explain this concept (rows 212-214).
11- When discussing the choice of laser technology, consider addressing the Technology Readiness Level (TRL) and any necessary advancements or research required to increase its TRL.
The TRL of the laser for space applications is very low, but research is becoming increasingly interested in this area, a sentence was added in the conclusion. (rows 585-586).
12- Consider providing more information on the overall efficiency and feasibility of the proposed system.
Added to conclusions (rows 590-594).
13- What specific developments or improvements related to this research can be expected in the future, and how will they enhance the system's performance?
Added a paragraph in the conclusions explaining how we can increase the energy sent to lunar soil mainly either by developing lasers with better efficiencies or by increasing the number of satellites (rows 602-606).
Comments on the Quality of English Language
There are a few areas where the language could be improved for better clarity and flow. Some sentences are quite long and complex, which could be broken down into smaller sentences for easier understanding. Additionally, there are a few instances of repetition and awkward phrasing that could be refined. Overall, with some minor editing and improvements, the English writing of the paper can be further enhanced.
We revised the English of the entire document, trying to break longer, more articulate sentences into shorter, simpler ones and making constructions more understandable.
